# Agroindustrial Wastes as a Support for the Immobilization of Lipase from *Thermomyces lanuginosus*: Synthesis of Hexyl Laurate

**DOI:** 10.3390/biom11030445

**Published:** 2021-03-17

**Authors:** Regiane K. de S. Lira, Rochele T. Zardini, Marcela C. C. de Carvalho, Robert Wojcieszak, Selma G. F. Leite, Ivaldo Itabaiana

**Affiliations:** 1School of Chemistry, Department of Biochemical Engineering, Federal University of Rio de Janeiro, Rio de Janeiro 21941-909, Brazil; regianekessias@eq.ufrj.br (R.K.d.S.L.); rochelezar@gmail.com (R.T.Z.); marcela.caetano18@gmail.com (M.C.C.d.C.); selma@eq.ufrj.br (S.G.F.L.); 2Univ. Lille, CNRS, Centrale Lille, University Artois, UMR 8181-UCCS-Unité de Catalyse et Chimie du Solide, F-59000 Lille France; robert.wojcieszak@univ-lille.fr

**Keywords:** lipases, *Thermomyces lanuginosus*, agroindustrial waste, hexyl laurate, lipase immobilization, biomass valorization, green chemistry, biocatalysis

## Abstract

As a consequence of intense industrialization in the last few decades, the amount of agro-industrial wastes has increasing, where new forms of valorization are crucial. In this work, five residual biomasses from Maranhão (Brazil) were investigated as supports for immobilization of lipase from *Thermomyces lanuginosus* (TLL). The new biocatalysts BM-TLL (babaçu mesocarp) and RH-TLL (rice husk) showed immobilization efficiencies >98% and hydrolytic activities of 5.331 U g^−1^ and 4.608 U g^−1^, respectively, against 142 U g^−1^ by Lipozyme^®^ TL IM. High esterification activities were also found, with 141.4 U g^−1^ and 396.4 U g^−1^ from BM-TLL and RH-TLL, respectively, against 113.5 U g^−1^ by TL IM. Results of porosimetry, SEM, and BET demonstrated BM and RH supports are mesoporous materials with large hydrophobic area, allowing a mixture of hydrophobic adsorption and confinement, resulting in hyperactivation of TLL. These biocatalysts were applied in the production of hexyl laurate, where RH-TLL was able to generate 94% conversion in 4 h. Desorption with Triton X-100 and NaCl confirmed that new biocatalysts were more efficient with 5 times less protein than commercial TL IM. All results demonstrated that residual biomass was able to produce robust and stable biocatalysts containing immobilized TLL with better results than commercial preparations.

## 1. Introduction

Due to the increase in the life expectancy, the world population has been growing alarmingly, bringing as immediate consequences the increase in industrial and agricultural activities as a way to supply the energy and food demands [1]. Besides, the unsustainability of fossil fuels requires a shift towards the use of renewable sources to meet future energy and chemical needs as a way to balance economic and environmental aspects through reducing emissions of greenhouse gases which cause climate change, global warming, and health issues, severely threatening the current environment [2].

As a direct consequence of this development, around 200 million tons of agro-industrial wastes has been generated worldwide, alarming the concern about the environmental impact caused and the development of new approaches for waste valorization [3]. In this trend, the integration between technologies developed by researchers and industry has been crucial to the search for short-term solutions where the reuse of waste and the search for new (bio)catalysts are essential to obtain more selective processes with higher yields in biorefineries in the future [4,5,6]. To this purpose, the only solution to effectively resolve the challenge is to discover advanced technologies for the sustainable development and production from eco-friendly biomass feedstock [7].

Biomass is currently the largest source of renewable carbon-based energy. Lignocellulosic biomass, consisting of cellulose (40–50%), hemicellulose (15–25%), and lignin (15–30%) is the most abundant source of carbon-based molecules [8]. This fact is often overlooked, as most of this biomass is used non-commercially for energy and heating, which is inefficient in terms of productivity, in addition to generating more pollution. Due to its chemical and structural complexity, biomass can be used in a better way through the synthesis of bio-based fuels or value-added chemicals [9,10].

Brazil has one of the greatest biodiversity on the planet, where agricultural extraction is highly economically important. Among the states that stand out in this activity, Maranhão is responsible for supplying more than half of the Northeast, North, and Southeast regions of the country due to climatic and soil conditions conducive to the cultivation of several agricultural genres [11]. As a result of this extractive activity, a high range of lignocellulosic by-products with a high concentration of organic material are generated, where many of them are still discarded without appropriate treatment in the soil or directed to serve as animal feed, destinations that a priori do not generate considerable economic advantages for the industry, besides representing logistical and environmental issues [12,13,14]. In this context, a better valorization of lignocellulosic biomass can be an auspicious source of new added value compounds, as it is abundant and diversified. In addition, due to its varied morphologies and appreciable hydrophobicity and rigidity, lignocellulosic biomass can also be applied as a support for the immobilization of enzymes, making it a renewable, abundant, and low-cost support for obtaining new biocatalysts [15,16]. Among the enzymes with notable industrial importance, lipases (triaciglycerol hydrolases, E.C. 3.1.1.3) have received great prominence in recent years [17]. Although the natural reaction of this class of enzymes is the hydrolysis of triacylglycerols, lipases are also capable of catalyzing synthesis reactions by recognition a high range of substrates, showing activity in organic solvents, in addition to acting with high regio-, enantio-, and chemo selectivity, factors that, together with the absence of co-factors, make these enzymes highly attractive in several industrial segments [18,19,20,21,22]. However, the applications of lipases in industrial processes are often hindered due to their low long-term stability and gradual decrease of activity during storage, expensive and/or inefficient recovery, and subsequent reuse [23]. These limitations can be overcome by applying appropriate immobilization techniques to improve their stability against denaturing agents and proteolysis, facilitate their recovery from the reaction media at the end of the processes for subsequent reuse with concomitant reduction of costs in downstream processes, avoid enzyme aggregation, develop processes with multienzymatic cascade reactions, and increase the flexibility of reactor designs [24,25]. The search for supports able to confer important interactions with enzymes and provide robust processes is still a field of intense study. However, the high cost of this technology still limits the application of immobilized enzymes on an industrial scale. In this context, lignocellulosic biomass, being abundant and multifunctional, can be an interesting and accessible alternative as a support for enzymatic immobilization [26,27,28,29].

In this work, residual biomass from sugarcane bagasse (SC), rice husks (RH), corn cobs (CC), babassu mesocarpus (BM), and coffee grounds (CG) from the industrial from Maranhão (Brazil) were investigated as supports for immobilization of the commercial lipase of *Thermomyces lanuginosus* (TL 100L-Novozymes^®^) with subsequent physico-chemical biocatalysts characterization and application in the hexyl laurate biosynthesis.

## 2. Materials and Methods

### 2.1. Materials

Lignocellulosic wastes from sugarcane bagasse (SC), rice husks (RH), corn cobs (CC), babassu mesocarpus (BM), coconut bark (CB), and coffee grounds (CG) used as support were collected in supply and processing centers in the State of Maranhão (Brazil). Free lipase from *Thermomyces lanuginosus* (Lipozyme TL100L-Novozymes^®^) with 22.8 mg. mL^−1^ of protein and respective corresponding immobilized on silicate support (Lipozyme TL IM-Novozymes^®^) were purchased by LNF-LatinoAmericana (Bento Gonçalves, Brazil). Bovine serum albumin (BSA), Comassie Brilliant Blue G250 dye, ethyl alcohol anhydrous, 95% and orthophosphoric acid 85%, oleic acid, acetone and, lauric acid, and n-hexanol were used in the enzymatic activity and protein quantification stages, and the Pink Cane dye was purchased from Sigma-Aldrich (St. Louis, MO, USA). Arabic Gum and olive oil (Gallo^®^) with low acidity used as a substrate in the hydrolytic activity test were purchased from Synth (São Paulo, SP, Brazil). All reagents used were of analytical grade.

### 2.2. Production of Biocatalysts

#### 2.2.1. Preparation of Supports

Initially, in-natura agro-industrial residues of coffee grounds (CG), corncobs (CC), sugar cane (SC), babassu mesocarp (BM), and rice husks (RH) were washed with distilled water, dried for 24 h at greenhouse (Figure 1a), crushed, sieved (28–35 mesh), and then washed with ethanol (99.8% PA) at room temperature [30,31]. Approximately 10 g of each washed support was subjected to degreasing in a Soxhlet extractor for 4 h with 200 mL of ethanol at 50 °C. After extraction, the sample present in the extraction cartridge was removed and dried in an oven at 60 °C for 6 h (Figure 1b). The defatted biomasses were applied in the immobilization protocols.

#### 2.2.2. Immobilization of TLL Lipase

TLL was immobilized by adsorption to the hydrophobic surface on defatted biomasses according to the methodology described in [32]: 1 g of each support was added to 10 mL of *n*-hexane and stirred at room temperature for 2 h. Then, 10 mL of enzymatic solution (containing 5 mg of protein per g of support) in sodium phosphate buffer solution pH 7.0 (0.025 or 0.1 M) was added to the medium under stirring for more 2 h following by conservation under static conditions at 4 °C for 3 to 24 h, where aliquots of the supernatant were taken in order to quantify the immobilization efficiency through the concentration of proteins and residual hydrolytic activity. At the end of the process, the resulting biocatalysts were filtered under vacuum and washed three times with 10 mL of *n*-hexane.

The efficiency of the adsorption process (*E_ads_*) was determined by Equation (1) [24]:(1)$$E_{ads}=\frac{C_{0}\;-\;C_{f}\;}{\;C_{0}}*100$$
where *C*_0_ and *C_f_* are the initial and residual protein concentration in the immobilization supernatant (mg·mL^−1^), respectively.

The lipase activation ratio in the new immobilized biocatalysts in terms of activity (*N_act_*) was determined by Equation (2) [33]:(2)$$N_{act}\;=\left(\frac{A_{i}}{A_{0}}\right)*100$$
where *A_i_* is the specific activity of the immobilized biocatalyst (U·g^−1^) on hydrolitc activity assay and *A_0_* is the hydrolytic activity offered at the beginning of the immobilization (15.90 U·g ^−1^).

#### 2.2.3. Determination of Protein Concentration

The concentration of proteins from the enzyme extract of free TLL and supernatants during the immobilization process was carried out according to the method of Bradford where the protein content was estimated by the average of a calibration curve obtained using albumin bovine serum (BSA) as standard [34].

#### 2.2.4. Determination of Hydrolytic Activity

The activities of free and immobilized lipases were determined as follows [28]: 1 mL of extract and 10 mg of each supported enzyme were added to 19 mL of an emulsion prepared with olive oil (5% *w*/*v*) and Arabic gum (10% *w*/*v*) in sodium phosphate buffer (100 mM, pH 7.0). The reactions were carried out under agitation (200 rpm) at 35 °C for 30 min. The reactions were then stopped by the addition of 20 mL of the acetone−ethanol mixture (1:1 *v*/*v*), and the fatty acids produced were extracted under agitation (200 rpm) for 10 min and titrated until end point (pH 11.0) with NaOH solution (0.04 N). The blank assays were performed by adding the extract just after the addition of the acetone−ethanol solution to the flask. One unit of lipase activity (U) was defined as the amount of enzyme that catalyzes the release of 1 μmol of fatty acids per minute, under the assay conditions.

#### 2.2.5. Determination of Esterification Activity

The esterification activity (U·g^−1^) of the new biocatalysts and the commercial lipase TL IM (positive control) was quantified through the reaction between oleic acid and ethanol in *n*-heptane (1: 1, 100 mM) [35,36]. The reaction was carried out at 40 °C, 160 rpm, and 5% (*w*/*v*) of biocatalyst for 40 min, followed by the addition of 20 mL of acetone–ethanol for the extraction of residual fatty acids. The amount of acid consumed was determined by titration with 0.04 M NaOH and using phenolphthalein as an indicator. One unit of activity was defined as the amount of enzyme that consumes 1 µmol of oleic acid per minute and calculated using Equation (3) [37]:(3)$$EA=\frac{\left(V_{b}\;-\;V_{a}\right)\;\times \;M\;\times \;V_{f}\;\times \;1000}{t\;\times \;m\;\times \;V_{C}}$$
where *V_a_* is the volume of NaOH titrated in the sample (mL); *V_b_* is the volume of NaOH titrated in the control sample (mL), M is the molar concentration of NaOH solution (mol. L^−1^), *V_f_* is the final volume of reaction medium (mL), *t* is the reaction time (min); *m* is corresponding to the mass of the sample (g of derivative or immobilized commercial enzyme), and *V_c_* is the volume of the aliquot removed from the reaction medium (mL).

### 2.3. Hexyl Laurate Synthesis

The new biocatalysts were applied in the synthesis of hexyl laurate (Scheme 1) through the esterification of lauric acid n-hexanol in *n*-hexane. The amount of each biocatalyst was adjusted to 100 U based on the data obtained from the hydrolysis assay. Solutions (10 mL) of lauric acid in n-hexanol/heptane (1:1, 0.1 M) were added to 25 mL conical tubes in the presence of the biocatalysts and stirred under for 2 h at 200 rpm and 37 °C in shaker.

Aliquots of 20 μL were collected from 5 to 120 min to study the time course reaction. All analyses were done in triplicate. The samples were derivatized with 20 μL of *N*-methyl-*N*-(trimethylsilyl)trifluoroacetamide (MSTFA) and graded to 1.0 mL of heptane. Subsequently, the samples were analyzed in a gas chromatography unit equipped with a mass spectrometry detector (GC−MS). Analyses were performed in a GC−MS system (Shimadzu CG2010, DB 5 capillary column). Samples were prepared by dissolving 10 μL of the final product in 980 μL of heptane and 10 μL of N-methyl-*N*-(trimethylsilyl)trifluoroacetamide (MSTFA). The injector and detector temperatures were 250 °C, and the oven temperature was constant at 60 °C for 1 min and then increased by 10 °C per min to 250 °C, when it was held constant for 3 min.

### 2.4. FT-IR Analysis

Confirmation of hexyl laurate in the best reaction condition was also qualitatively determined by Fourier transform infrared spectroscopy (FTIR), on a Thermo-Nicolet Magna^®^ spectrophotometer (IR 760), in the 400–4000 cm^−1^ spectral range. The experiments consisted of 200 readings with a resolution of 2 cm^−1^.

### 2.5. Support Desorption

The new biocatalysts were subjected to protein desorption assays according to the methodology previously described [38,39]: 1 g of immobilized biocatalyst was poured into 50 mL of 25 mM phosphate buffer, pH 7.0, containing 2.5% (*v*/*v*) of Triton X-100 or NaCl 0.8 % (*w*/*v*). The supports immersed in the desorption solution were kept under constant stirring at 100 rpm for 4 h at room temperature. At the end of this period, the particles were separated from the supernatant (vacuum filtered), the supernatant being subjected to protein quantification assay, and the particulate material applied in the hexyl laurate esterification reaction.

## 3. Results and Discussion

### 3.1. Investigation of TLL Adsorption as a Function of Ionic Strength and Time

For the construction of a robust biocatalyst, the selection of the appropriate immobilization method is a crucial step, and it is related to the knowledge of the properties both of enzyme and supports [40]. In this work, we proposed a TLL immobilization strategy based on the adsorption on the surface of the previously degreased residual biomass from Maranhão, avoiding further functionalization or derivatization steps. Initially, the influence of the ionic strength of sodium phosphate buffer pH 7.0 in two concentrations was investigated: 0.025 M (low ionic strength) and 0.1 M (high ionic strength) on the adsorption efficiency (E_ads_) of TLL in the different supports, the results of which are shown in Figure 2.

In general, the highest E_ads_ was found in the lowest ionic strength, where the best value was obtained for the SB derivative with 82.24% in 0.025 M and 72.54% in 0.1 M phosphate buffer. At low ionic strengths, lipase is solvated with the lowest ion charge possible, enabling greater hydrophobic interaction with the support, without protein aggregation [41], as at neutral pH, the amino acids of TLL surfaces are mostly at nonionized form [42,43]. In order to observe the effects of immobilization at low ionic strength over time, all defatted biomasses were subjected to a 24 h kinetic test, in order to find the shortest immobilization time (Figure 3).

As a result, a saturation of TLL immobilization in 9 h could be observed by all supports, with E_ads_ > 98%. This immobilization time is much lower than that reported by [44] when immobilizing lipase of *Candida rugosa* by adsorption, where 28 h of immobilization were necessary. This contact time was also reported by [45], when immobilizing the lipase of *Burkholderia cepacia* on pretreated corn cob. Such results show a high affinity between enzyme and support, and therefore 9 h of immobilization and 0.025 M of ionic strength were adopted for subsequent studies.

### 3.2. Selection of Biocatalyst

In order to choose the best immobilized biocatalyst for hexyl laurate synthesis, the activities of hydrolysis and esterification in addition to the enzyme activation index in the final derivatives were determined (Table 1).

As a result, the immobilized biocatalysts BM-TLL, RH-TLL, and CG-TLL presented the most promising hydrolytic activities, where 5331 U g^−1^ was obtained by BM-TLL, which corresponds to an activation index of 335% of the TLL taking into account the initial activity offered of free TLL before the immobilization process (318.5 U g^−1^). For RH-TLL and CG-TLL, high activation rates were also obtained, being 290 and 108%, respectively. Note that all hydrolytic activities found were significantly superior to those found by the commercial biocatalyst TL IM (142.3 U g^−1^), which reveals promising results.

The high results of hydrolytic activity obtained by the new derivatives can be justified by a possible hyperactivation of TLL in the presence of hydrophobic surfaces of the support [46,47]. In the absence of hydrophobic surfaces or emulsions, the TLL has its conformational balance shifted to the closed and inactive form, where a hydrophobic lid containing an alpha-helical polypeptide chain covers the active site, hindering its catalytic activity [48]. However, in the presence of hydrophobic interfaces, such as the residual biomasses applied in this work, the open conformation of the lid is favored and stabilized by several hydrophobic interactions of its outer face with the support during the adsorption process, generating distortions, and exposure of the catalytic site, increasing intensively the catalytic activity [49,50]. Our results were superior of those obtained in [51,52,53], whose previously reported the immobilization and stabilization of TLL in several different organic supports.

The esterification activity of the new biocatalysts was also investigated through ethyl oleate synthesis (Table 1). Surprisingly, the biocatalysts CG-TLL and SC-TLL did not show any esterification activity. Although they have shown high hydrolysis activity, as porous supports, other interactions between enzyme and biomass may have caused diffusion limitations, where the enzyme can be stocked in compartments of difficult access, limiting the substrate access [54]. In addition, as TLL requires interfacial activation, this phenomenon is disturbed in the presence of organic solvents, which makes the lipase structure more rigid and hinders its agility, resulting in lower esterification yields, as reported in previous studies by [48,55]. On the other hand, BM-TLL demonstrated esterification activity slightly higher than commercial TLIM, with 141.4 U g^−1^ against 113.5 U g^−1^ for commercial preparation. CC-TLL showed esterification activity intensely lower than TL IM (26 U g^−1^), while RH-TLL presented the best results, with 396 U g^−1^.

These promising results must also be explained by taking into account the previously performed physical-chemical characterizations of defatted supports before immobilization. The degreasing process leads to the formation of new pores in the materials, forming new potential points of interaction and enzyme entrapment. In the elementary analysis for determination of C, N, H, and S, as well as the determination of cellulose, hemicellulose, and lignin content, the lignocellulosic nature of the materials studied was confirmed (Appendix A), which are in accordance with those already reported in the literature [56,57]. The high amount of carbon demonstrated in addition to the percentage of lignin denoted a high hydrophobic surface, which together with the large number of pores in the materials, favored multiple interactions with the enzyme.

The consequences of the comminution and degreasing process can be seen in Figure 4, which shows the SEM analysis performed with each defatted support, where it is possible to observe the irregular and porous nature of these materials, characteristic of lignocellulosic residues. For this reason, TLL probably was immobilized on these supports not only by adsorption to the surface, but also by confinement in the pores of these materials, characterizing multiple interactions.

Due to be heterogeneous supports, other physical-chemical characteristics need to be analyzed. Table 2 shows the values found for hydrophobicity by the fixation test of the cane pink dye, in addition to the measurements of the surface area, volume, and pore diameter of the defatted supports.

Support hydrophobicity is an important property to be considered when immobilizing TLL, once interfacial activation is necessary due to the presence of a hydrophobic lid. In these cases, immobilization by surface adsorption is an interesting strategy to immobilize the lipase in its open conformation [46,47,58]. According to Table 2, the RH support showed the lowest hydrophobicity (724 μg g^−1^), followed by BM, SG, CC, and CG. However, in heterogeneous supports, the interaction with the enzyme is not only carried out in a hydrophobic way, as the greatest hydrolysis activity was found by the biocatalyst BM-TLL (5331 U g^−1^, Table 1), whose support presented the second lowest hydrophobicity (Table 2). In addition to hydrophobicity, the surface area, volume, and the pore diameter are characteristics of particular importance on lipase immobilization. The surface area of the studied supports obtained by the BET method varied from 0.3 to 1.4 m^2^ g^−1^. This parameter reveals the actual specific surface area available for the interaction with the enzyme [59]. In this sense, RH support proved to be quite interesting, as its surface area (1.42 m^2^ g^−1^) was approximately seven times greater than the value found for SC (0.30 m^2^ g^−1^), and twice as much the surface area of the CC support (0.79 m^2^ g^−1^). QI et al. (2015) [60] when applied defatted and chemically modified rice bran as supports in the immobilization of lipases, observed values much lower than those reported in this work, obtaining 0.68 m^2^ g^−1^ and 0.67 m^2^ g^−1^, respectively. In addition, RH was the support with the largest pore volume (23.21 × 10^−4^ cm^3^ g^−1^). The pore volume of a support can influences the immobilization process, as it can limit the amount of enzyme that can be lodged inside it or direct the protein adsorption for only on the surface [61]. Regarding the pore diameter, the values obtained ranged from 11.6 to 65.4 Å, with the SC (11.7 Å) and CC (19.3 Å) supports classified as microporous and CG, BM and RH classified as mesoporous [62]. Porous supports can be responsible for hosting large amounts of protein, influencing catalytic properties of biocatalysts, such as immobilization yield, selectivity and specificity, and molecular diffusion (substrates and/or products) in the biocatalyst microenvironment [26,63]. In addition, the TLL structure has been described as of spherical shape and diameter of 35 × 45 × 50 Å, that is ideal for immobilization on supports containing pores within the range of mesopores (20–500 Å) [48]. Thus, the biocatalysts BM-TLL and RH-TLL, which presented the best results of hydrolysis and esterification activities, were able to form a set of important interactions that turn TLL more active and stable under the conditions studied, combining the hydrophobic and porous effects in the activation and enzyme entrapment. These were selected for the application stage in the synthesis of hexyl laurate.

### 3.3. Synthesis of Hexyl Laurate

In this step, the biocatalysts BM-TLL and RH-TLL were submitted to a kinetic study for the formation of hexyl laurate in *n*-hexane (Figure 5). As a result, the RH biocatalyst showed conversions of 65.3% in 4 h of reaction against 94.2% demonstrated by BM-TLL. When comparing these results to those obtained in [64] for the synthesis of hexyl laurate catalyzed by commercial TL IM, which reported a conversion of 92.2% using 45.5% (*w*/*v*) of commercial biocatalyst, the results presented in this work are promising as it was obtained higher conversions with only approximately 5 mg of immobilized enzyme per g of support, 5% (*w*/*v*) of BM-TLL (Table 1) in order to contemplate 100 U of standardized activity.

Analysis by GC-MS (Appendix A) and FT-IR spectra on KBR disk (Appendix A) confirmed high selectivity of BM-TLL for hexyl laurate formation as well as purity of starting materials.

### 3.4. Desorption of TLL by BM-TLL Biocatalyst

In order to confirm the type of the predominant interaction in the BM-TLL biocatalyst, desorption tests with Triton X-100 and NaCl were performed. The results were compared with the commercial enzyme TL IM (Table 3).

Triton X-100 and NaCl were applied in order to break the hydrophobic and electrostatic interactions, respectively. As noted, almost all the protein present in the BM-TLL biocatalyst was adsorbed by Triton X-100, which competes with lipases for the hydrophobic regions of the support, causing desorption by molecular exclusion [58]. In addition, the amount of protein detected in the supernatant corroborates the data previously obtained on immobilization procedures. The commercial lipase TLIM presented the highest percentage of desorbed proteins when treated with NaCl, which reduces the negative charges present in the support, resulting in less ionic interaction between the lipase molecules and the support, corroborating an immobilization through ionic adsorption, differently interactions found in the BM-TLL biocatalyst. In addition, these studies demonstrated a quantity of proteins in TLIM that was much higher than that found by the home-made biocatalyst, demonstrating that the enzyme immobilized in the lignocellulosic residue did indeed present intense hyperactivation and stability, as it presented superior catalytic activities with lower amounts of protein.

The reusability of the new biocatalyst BM-TLL was tested in subsequent cycles of hexyl laurate synthesis (Figure 6), where after approximately 11 cycles, the enzyme continued to show the same conversion values.

Subsequent conversion drops were observed, where 50% conversion was detected after 25 recycles. These results may be a consequence of the escape of proteins into the reaction medium as a result of the degradation of the support with the advance of the cycles of use of the biocatalyst. This effect can be highlighted on Figure 7 that shows the SEM analysis of the BM-TLL biocatalyst before (Figure 7a) and after (Figure 7b) the reaction cycles. It is possible to observe that after the recycles, the surface has undergone small changes, making it possible to more easily observe the presence of proteins, probably due to the escape effect. Even so, these results show high robustness and stability of homemade immobilized enzyme.

## 4. Conclusions

In this work, we demonstrated the application of five residual biomasses from Maranhão (Brazil) as supports for TLL immobilization as a way of valorization of these wastes. The babaçu mesocarp (BM) offered the best physical-chemical characteristics for TLL hydrophobic adsorption and entrapment, resulting in a competitive homemade biocatalyst (BM-TLL) that showed higher hydrolytic and esterification activities than the commercial Lipozyme^®^ TL IM with 4 times lower protein amounts. On hexyl laurate synthesis, BM-TLL it was possible to acquire 94% of conversion in 4 h of reaction and with high recyclability of the homemade biocatalyst, demonstrating to be a robust and renewable alternative to commercial biocatalyst.

## Data Availability

The data presented in this study are available in this paper and Appendix A.

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
