# Peer review of "Agroindustrial Wastes as a Support for the Immobilization of Lipase from Thermomyces lanuginosus: Synthesis of Hexyl Laurate"

_biomolecules, 2021, doi:10.3390/biom11030445_

Round 1

Reviewer 1 Report

The manuscript entitled “Agroindustrial wastes as a support for the immobilization of lipase from Thermomyces lanuginosus: synthesis of hexyl laurate” reports on the use of different agroindustrial wastes for lipase immobilization, their evaluation and the choice of the most promising biocatalyst to be used for ester synthesis.

It is a very interesting work, that covers a very crucial subject, that of wastes exploitation. The authors have done a good planning and research design. The results presented support the conclusions. The language needs a check as errors are present, especially at the Introduction section. There are some points that need to be addressed. More specifically:

The title of the main manuscript is different than the title of the SI.

Authors’ affiliations are different on the main manuscript and on the SI.

Line 101: The authors describe 6 agroindustrial wastes to be examined, among which coconut bark. However, everywhere else in the manuscript they test, examine and present results on 5 agroindustrial wastes, which doesn’t include coconut bark.

Line 218: For the reaction conditions the authors give a temperature of 40 °C, while on the scheme that follows (Scheme 1) they give a temperature of 37 °C.

Line 287, Table 1: The authors give on this table the lipase activation index, Nact. The equation on which their calculations are based is given on Line 166 (equation 2). Taking into account equation 2, the values of the column “Hydrolytic activity” and the value of initial lipase activity given on Line 294 (being 318.5 U. g-1), the values given on column “Nact” are not correct. The authors should correct them.

Line 320: The authors claim that “…BM-TLL, CC-TLL and RH-TLL derivatives showed esterification activities far superior to the commercial preparation TL IM (113.5 U. g-1).” When BM-TLL gives an activity of 141.4 U. g-1 and CC-TLL gives 26 U.g-1 one cannot claim these values are far superior to 113.5. For BM-TLL the activity is slightly better, for CC-TLL is definitely lower. Besides, the authors recognize this, and they continue the synthetic experiments for BM-TLL and RH-TLL, only.

Line 327 “…elementary analysis…”: the authors should say what method was used for the analysis

Table 2 “hydrophobicity” column: there one can see for CG the number 12,03. A digit is missing. Is this 12,003 or 12,030 or something else? Besides, the numbers on this column are not presented in the same way. E.g. 2600 should be 2,600 and 1298 should be 1,298 - to be similar with 14,782 below them.

The same should apply to all numbers on both tables. The number of decimals should also be the same where the are decimals.

Line 370 “…varied from 0.2 to 1.4”: What material presented 0.2? It is not presented on Table 2.

Lines 381-382: “…with the SC (11.670 Å) and CC (19.318 Å)” the authors should be more careful with the use of decimals.

Lines 410-411 “Analysis by GC-MS (Figure S1 of supporting information) and FT-IR spectra on KBR disk (Figure S2 of SI)…” : GC-MS analysis is missing from SI. Fig. SI 1 is the FTIR spectrum. Moreover, the figure caption is in Portuguese.

After these minor revisions and performing a spell check, I suggest accepting the manuscript for publication.

Author Response

Answers To the Referees
Referee 1

The manuscript entitled “Agroindustrial wastes as a support for the immobilization of lipase from Thermomyces lanuginosus: synthesis of hexyl laurate” reports on the use of different agroindustrial wastes for lipase immobilization, their evaluation and the choice of the most promising biocatalyst to be used for ester synthesis.

It is a very interesting work, that covers a very crucial subject, that of wastes exploitation. The authors have done a good planning and research design. The results presented support the conclusions. The language needs a check as errors are present, especially at the Introduction section. There are some points that need to be addressed. More specifically:

The title of the main manuscript is different than the title of the SI.

Answer: Dear referee, thank you for your observation. It was correctly changed.

Authors’ affiliations are different on the main manuscript and on the SI.

Answer: Dear referee, thank you for this remark. It was also changed.

Line 101: The authors describe 6 agroindustrial wastes to be examined, among which coconut bark. However, everywhere else in the manuscript they test, examine and present results on 5 agroindustrial wastes, which doesn’t include coconut bark.

Answer: Dear referee, thank you for your careful observation. The coconut shell was initially planned to be part of this work, but has not been investigated. For this reason, we have no results to show. This biomass was then excluded from the text.

Line 218: For the reaction conditions the authors give a temperature of 40 °C, while on the scheme that follows (Scheme 1) they give a temperature of 37 °C.

Answer: Dear referee, thank you for your careful observation. Indeed, the correct temperature was 37°C. This data was changed in the text

Line 287, Table 1: The authors give on this table the lipase activation index, Nact. The equation on which their calculations are based is given on Line 166 (equation 2). Taking into account equation 2, the values of the column “Hydrolytic activity” and the value of initial lipase activity given on Line 294 (being 318.5 U. g-1), the values given on column “Nact” are not correct. The authors should correct them.

Answer: Dear referee, thank you for the comment. The lipase activation index, Nact was calculated taking into account the final hydrolytic activity obtained by the new immobilized biocatalyst, and the hydrolytic activity of the TL100L lipase initially offered, in relation to the amount of support. Our mistake was not to mention the initial activity of TL 100L, which was 15.90 U. g-1. Using this value, and not those obtained by commercial lipase TLIM, the values ​​obtained are the ones we mentioned in the table. This value (15.90 U. g-1) was added in the methodology and in the footnote to the table 1.

Line 320: The authors claim that “…BM-TLL, CC-TLL and RH-TLL derivatives showed esterification activities far superior to the commercial preparation TL IM (113.5 U. g-1).” When BM-TLL gives an activity of 141.4 U. g-1 and CC-TLL gives 26 U.g-1 one cannot claim these values are far superior to 113.5. For BM-TLL the activity is slightly better, for CC-TLL is definitely lower. Besides, the authors recognize this, and they continue the synthetic experiments for BM-TLL and RH-TLL, only.

Answer: Dear referee, thank you for this observation. These commentaries were carefully changed in the manuscript

Line 327 “…elementary analysis…”: the authors should say what method was used for the analysis

Answer: Dear referee, thank you for this correction. The required information was added in the body of the text.

Table 2 “hydrophobicity” column: there one can see for CG the number 12,03. A digit is missing. Is this 12,003 or 12,030 or something else? Besides, the numbers on this column are not presented in the same way. E.g. 2600 should be 2,600 and 1298 should be 1,298 - to be similar with 14,782 below them.The same should apply to all numbers on both tables. The number of decimals should also be the same where the are decimals.

Answer: Dear referee, thank you for this observation. The table was corrected.

Line 370 “…varied from 0.2 to 1.4”: What material presented 0.2? It is not presented on Table 2.

Answer: Dear referee, thank you for this careful observation. Indeed, the correct initial value is 0.3. We corrected this sentence in the text.

Lines 381-382: “…with the SC (11.670 Å) and CC (19.318 Å)” the authors should be more careful with the use of decimals.

Answer: Dear referee, thank you for this careful observation. It was correctly changed in the text.

Lines 410-411 “Analysis by GC-MS (Figure S1 of supporting information) and FT-IR spectra on KBR disk (Figure S2 of SI)…” : GC-MS analysis is missing from SI. Fig. SI 1 is the FTIR spectrum. Moreover, the figure caption is in Portuguese.

Answer: Dear referee, thank you for your remark. The SI and manuscript was carefully changed.

After these minor revisions and performing a spell check, I suggest accepting the manuscript for publication.

Reviewer 2 Report

Dear Authors,

I would like to congratulate on the novelty of the idea of using wste biomass materials to immobilize the enzymatic protein. Moreover, the manuscript has been well written and the results support formulated conclusions.

I would reccommend some minor changes:

  • in the Supplementary Materials please write the titles in Figure  in English,
  • in the Supplementary Materials please change commas into dots in table 1, write the explanation of the abbreviation (CG, SC), write significant numbers of standard deviations,
  • line 198 - write n i n-heptane in italics,
  • figure 4 - the quality of the photographs ccould be improved,
  • figure 8 - did you tested non-immobilized lipase?

Author Response

Answers To the Referees
Referee 2

Dear Authors,

I would like to congratulate on the novelty of the idea of using wste biomass materials to immobilize the enzymatic protein. Moreover, the manuscript has been well written and the results support formulated conclusions.

I would reccommend some minor changes:

  • in the Supplementary Materials please write the titles in Figure  in English.

Answer: Dear Referee, thank you for corrections: The figures was changed

  • in the Supplementary Materials please change commas into dots in table 1, write the explanation of the abbreviation (CG, SC), write significant numbers of standard deviations,
  • Answer: Dear Referee, thank you for your suggestion: The table was changed.

  • line 198 - write n i n-heptane in italics,

Answer: Dear Referee, thank you for your observation. It was corrected.

  • figure 4 - the quality of the photographs ccould be improved.

Answer: Dear Referee, thank you for your suggestion. We improved the quality of figure 4

  • figure 8 - did you tested non-immobilized lipase?

Answer: Dear Referee, thank you for your question. The enzymatic preparation TL 100L, the non-immobilized enzyme, was tested but did not show satisfactory conversion results. Probably, because it is an enzyme in liquid and aqueous phase, there was not contact surface enough for the catalysis, since the solvent used in the reaction was n-hexane. As figure 8 was intended to compare immobilized biocatalysts, we decided not to add this data.

Reviewer 3 Report

The authors provided their original research paper dealing with immobilization of lipase from Thermomyces lanuginosus using agroindustrial wastes, such as sugarcane bagasse, rice husks, corn cobs, babassu mesocarpus, coconut bark and coffee grounds. In my opinion, the manuscript is well written. Materials and methods were described in correct and concise way. Introduction provided sufficient background and interesting data referring to e.g. lignocellulosic by-products. Results are promising due to the possibility of using agroindustrial wastes which are abundant worldwide, as well as, high yield of conversion of hexyl laurate. In my humble opinion, the submitted manuscript should be accepted after minor revision.

  • The title in the manuscript and in the supplementary materials is not the same.
  • n in n-hexane, as well as, N in N-Methyl-N-(trimethylsilyl)trifluoroacetamide should be written in italics. It should be improved throughout the manuscript.
  • The materials and methods part lacks the description of statistical analysis. Are you able to perform post-hoc tests to evaluate whether the results differ statistically (e.g. in Figure 2 or Table 1)?
  • Page 8, line 327 – change “table 1” to “table S1”
  • Please improve English in the manuscript, e.g. "hidrofobicity"

Moreover, in the Supplementary materials:

  • Figure S1 should be translated into English.
  • Table S1 – support names should be defined in the table footer.

Author Response

Answers To the Referees
Referee 3

The title in the manuscript and in the supplementary materials is not the same.

Answer: Dear referee, thank you for your observation. The title was changed.

n in n-hexane, as well as, N in N-Methyl-N-(trimethylsilyl)trifluoroacetamide should be written in italics. It should be improved throughout the manuscript.

Answer: Dear referee, thank you for your observation. These issues were solved.

The materials and methods part lacks the description of statistical analysis. Are you able to perform post-hoc tests to evaluate whether the results differ statistically (e.g. in Figure 2 or Table 1)?

Answer: Dear referee, thank you for the question. Although we are able to carry out statistical analysis, restrictions on access to the laboratory due to the covid-19 pandemic temporarily prevent us from accessing the research laboratories, where the software is installed. We apologize deeply.

Page 8, line 327 – change “table 1” to “table S1”

Answer: Dear referee, thank you for your observation. This issue was solved.

Please improve English in the manuscript, e.g. "hidrofobicity"

Answer: Dear referee, thank you for your suggestions. Some sentences and words were carefully changed in the manuscript.

Moreover, in the Supplementary materials:

Figure S1 should be translated into English.

Answer: Dear referee, thank you for your correction. It was corrected.

Table S1 – support names should be defined in the table footer.

Answer: Dear referee, thank you for your observation. This issue was solved.
